# Evaluation of the effectiveness of potassium chloride in the management of out-of hospital cardiac arrest by refractory ventricular fibrillation: Study protocol of the POTACREH study

**Romain Jouffroy**[1]*, **Patrick Ecollan**[2], **Charlotte Chollet-Xemard**[3], **Bertrand Prunet**[4], **Caroline Elie**[5], **Jean-Marc Treluyer**[5], **Benoit Vivien**[1]

1 SAMU de Paris, Service d'Anesthésie-Réanimation, Hôpital Universitaire Necker—Enfants Malades, APHP Centre, Assistance Publique—Hôpitaux de Paris and Université de Paris, Paris, France, 2 SMUR Pitié Salpêtrière, Groupe Hospitalier Pitié-Salpêtrière, Assistance Publique—Hôpitaux de Paris, Paris, France, 3 SAMU du Val de Marne, Hôpitaux Universitaires Henri Mondor, Assistance Publique—Hôpitaux de Paris, Créteil, France, 4 Brigade de Sapeurs-Pompiers de Paris, Paris, France, 5 URC Cochin, Assistance Publique—Hôpitaux de Paris, Paris, France

* romain.jouffroy@gmail.com

**Data Availability Statement:** No datasets were generated or analysed during the current study. All

## Abstract

### Purpose

Out-of-hospital cardiac arrest (OHCA) has a poor prognosis, with an overall survival rate of about 5% at discharge. Shockable rhythm cardiac arrests (ventricular fibrillation (VF) and pulseless ventricular tachycardia (VT)) have a better prognosis. In case of shockable rhythm, treatment is based on defibrillation, and thereafter, in case of failure of 3 external electric shocks (EES), on direct intravenous administration of 300 mg amiodarone, or lidocaine when amiodarone is unavailable or inefficient.

During surgical procedures under extracorporeal circulation, a high potassium cardioplegia solution is administered to interrupt cardiac activity and facilitate surgical procedure. By extension, direct intravenous administration of potassium chloride (KCl) has been shown to convert VF, resulting in return to a hemodynamically efficient organized heart rate within a few minutes.

The aim of this study is to provide clinical evidence that direct intravenous injection of KCl, into a patient presenting with OHCA due to refractory VF although 3 EES, should interrupt this VF and then allow rapid restoration of an organized heart rhythm, and thus return of spontaneous circulation (ROSC).

### Methods

A multicenter, prospective, single group, phase 2 study will be conducted on 81 patients presenting with refractory VF. After failure of 3 EES, each patient will receive direct intravenous injection of 20 mmol KCl instead of amiodarone. The primary outcome will be survival rate at

relevant data from this study will be made available upon study completion.

**Funding:** The authors received no specific funding for this work.

**Competing interests:** The authors have declared that no competing interests exist.

**Abbreviations:** OHCA, out-of-hospital cardiac arrest; VF, ventricular fibrillation; VT, ventricular tachycardia; KCl, potassium chlorine; CA, cardiac arrest; IV, intravenous; IVD, intravenous direct; ECC, extra corporeal circulation; EES, external electric shocks; ROSC, return of spontaneous circulation.

hospital admission. Major secondary outcomes will include ROSC and time to ROSC in the prehospital setting, number of VF recidivism after KCl injection, survival rate at hospital discharge with a good neurologic prognostic, and survival rate 3 months after hospital discharge with a good neurologic prognostic.

## Results

No patient is currently included in the study.

## Discussion

Conventional guideline strategy based on antiarrhythmic drug administration, i.e. amiodarone or lidocaine, for OHCA due to shockable rhythm, has not yet demonstrated an increase in survival at hospital admission or at hospital discharge. This may be related to the major cardiodepressant effect of those drugs.

## Trial registration

ClinicalTrials.gov Identifier: NCT04316611. Registered on March 2020. AP-HP180577 / N˚ EUDRACT: 2019-002544-24. Funded by the French Health Ministry. https://clinicaltrials. gov/ct2/show/NCT04316611.

## Purpose

Sudden cardiac arrest (CA) is defined as the sudden interruption of spontaneous circulation and represents a major public health problem by annually affecting more than 420,000 patients in the USA and between 30,000 and 50,000 in France [1]. About 85% of CA occur in the out-of-hospital settings [1]. The survival rate from out of hospital cardiac arrest (OHCA) is poor, nearby of 5% at hospital discharge [2]. However, the survival rate from OHCA related to "shockable" rhythm (ventricular fibrillation (VF) and pulseless ventricular tachycardia (VT)) is better, around 30% at hospital discharge [3].

Advanced life support (ALS) for OHCA related to "shockable" rhythm is based on external electric shock (EES), and after 3 unsuccessful EES, on anti-arrhythmic drug administration. Two drugs are classically recommended in this indication known as CA by "refractory" shockable rhythm: amiodarone (300 mg intravenously (IV)) as first line, and lidocaine (1 mg/kg IV) as second line [2,4,5]. Nevertheless, according to studies published in the literature, whereas these two drugs may increase the rate of return of spontaneous circulation (ROSC) and/or the rate of patients admitted alive in hospital, it should be noted that the results are much more mixed regarding survival at 3 months and/or at hospital discharge, which is the "strong" criterion generally used to assess survival after OHCA [6]. These negative results in terms of survival can probably be explained by the pharmacokinetic and pharmacodynamic properties of the two antiarrhythmic molecules, amiodarone and lidocaine. Amiodarone, which is a class III antiarrhythmic agent according to the Vaughan-Williams classification, has many side effects and in particular a bradycardizing effect and a negative dromotropic effect (slowing down of conduction), both more pronounced during high-dose administration and intravenously, which is the case during refractory OHCA. Lidocaine, which is a class Ib antiarrhythmic agent according to the Vaughan-Williams classification, induces a myocardial depressant effect, characterized by hypotension, myocardial depression and bradycardia. It is therefore perfectly possible that the beneficial effects of these 2 molecules in terms of immediate ROSC would be

counterbalanced by their negative cardiovascular side effects "persistent" after high dose IV administration, and thus ultimately do not allow any improvement in survival upon discharge from hospital. Consequently, it appears through the analysis of the literature that no molecule is more effective than a placebo in terms of survival at discharge from hospital in patients with OHCA due to VF or no-pulse VT, refractory to 3 EES. Other pharmacological research pathways are therefore necessary to try to improve the prognosis of these patients.

During surgical procedures performed under extracorporeal circulation (ECC), in particular cardiac valvular and/or coronary surgery, a cardioplegia solution is administered to interrupt the mechanical activity of the heart and thus facilitate the surgical procedure. Many cardioplegia solutions are available, varying in their chemical composition. Patho-physiologically, the essential element of these solutions is always represented by at a high concentration in potassium chloride, usually between 20 and 30 mmol/l [7]. The direct intravenous administration of this solution is then responsible for a transient hyperkalaemia, which has the immediate consequence of lowering the membrane resting potential of the myocytes, causing the cessation of their electrical and mechanical activity, and consequently inducing the immobility of the myocytes of the heart muscle. While these cardioplegia solutions are effective in stopping myocardial electrical activity during ECC implementation, some authors have by extension evaluated their effectiveness in interrupting VF. Since the first scientific publication by Weinstock and Clark in 1961 describing the administration of KCl in a 3-year-old child presenting with CA with refractory VF during an impossible intubation in the operating room, several cases series and human clinical studies have reported the efficacy and the safety of KCl injection to convert refractory VF or no-pulse VT [7–12]. However, most of these publications concerned patients during or after ECC for cardiac surgery requiring cardioplegia [8–12]. Elsewhere, we recently published the case of a resuscitation patient under ECC in intensive care unit, who presented with refractory VF after an OHCA: direct IV injection of 40 mmol of KCl allowed return to sinus rhythm within a few minutes, without subsequent recurrence of arrhythmias, and the patient was finally discharged alive from hospitalization with a good prognosis in terms of neurological function [13].

These various studies and clinical cases reporting the efficacy of a KCl injection are supported by experimental data. Indeed, it has been showed on an isolated endocardial fiber model that a concentration of 12 mM of KCl allows VF to give way, with a subsequent return to an organized electrical rhythm [14]. Elsewhere, a recent electrophysiological study on an isolated heart model has shown that hyperkalaemia acts mainly on the dynamic character and on the temporo-spatial organization of VF [15].

The question that obviously arises after an IV injection of KCl is the pharmacokinetics of the hyperkalaemia. Although studies are limited in this area, some data are nevertheless available in the literature. First, in their series of 100 patients who received an injection of 20 mmol of KCl, Øvrum et al. reported serum potassium values at the upper limit of normal values ($5.5 \pm 1.0$ mmol/l) 10 minutes after injection, and returned within normal values ($4.3 \pm 0.4$ mmol/l) after 20 minutes [9]. This therefore attests to the extremely transient nature of this hyperkalaemia following a direct IV injection of 20 mmol of KCl.

Finally, regarding the possible morbidity and mortality associated with an injection of potassium chloride, Almdahl et al. published reassuring data in a second article [16] using the same methodology as their first one [10]. In a series of 12,113 patients operated for cardiac surgery under ECC and cardioplegia, survival at D30 was not different between the 9,723 patients who did not present VF post-ECC and the 1,877 patients who presented VF successfully converted by an injection of 20 mmol of potassium chloride (1.2% vs 1.3%, p = 0.269). On the other hand, there was a tendency for an increase in long-term mortality in the 400 patients who presented VF resistant to "pharmacological defibrillation" and therefore required an

internal electric shock (hazard ratio = 1.19; 95% CI [0, 99–1.4], p = 0.07). These results are therefore reassuring with respect to possible deleterious consequences that could have been linked to the injection of KCl. They also suggest (but with all the limit of non-statistically significant results) that this pharmacological defibrillation could improve survival by comparison with conventional defibrillation performed by electric shocks (internal electric shocks in this study since it included patients operated for cardiac surgery).

All those works therefore suggest that a direct intravenous injection of 20 mmol of KCl in an OHCA patient presenting with VF or no-pulse VT, refractory to 3 electric shocks, thanks to a transient hyperkalaemia, could give way to this arrhythmia and subsequently allow a return to an organized and hemodynamically efficient heart rate, without any immediate or long-term deleterious consequences.

## Methods

### Justification of the study

This is a prospective non-comparative phase II clinical trial. The goal is to have an initial assessment of the effectiveness of KCl to interrupt refractory VF and allow ROSC. If the efficacy is found to be satisfactory (survival at admission close to 50%), a national randomized trial will be conducted to compare the strategy including KCl as the first line to the classic strategy.

In the absence of preliminary data on efficacy of KCl on OHCA due to refractory VF, it would not have been relevant to immediately propose a randomized trial against placebo. Indeed, in this context, to show a clinically significant improvement in survival, it would be necessary to include a large number of subjects (for example 1500 subjects per group to show an improvement in survival of 35% to 40%). Consecutively, the study we propose here is therefore an essential prerequisite.

### Intervention

Fig 1 presents the management algorithm for a patient with OHCA by VF according to international recommendations (Fig 1A) and in the context of this research (Fig 1B).

According to international guidelines, a direct administration of 300 mg IV of amiodarone should be performed in case of refractory (or recurrent) VF after 3 EES. If VF persists, a second administration of amiodarone IV should be given at half dose (150 mg) after the 5th EES. If the VF still persists, it is then possible to start continuous IV administration by electric syringe at a dose of 900 mg/day, or to resort to direct IV administration of lidocaine (1 mg/kg).

During this research, the injection of KCl 20 mmol will be performed after the 3rd electric shock, instead of the 1st administration of amiodarone. If necessary, the 1st injection of amiodarone 300 mg IVD will be performed after the 5th electric shock, and the 2nd injection 150 mg IVD after the 7th electric shock, and the relay with continuous amiodarone or the use of lidocaine afterwards.

### Population

**Study setting.** Out-of-hospital setting, all patients will be included by the mobile intensive care unit physicians.

### Inclusion criteria

- Adult patient (age 18 or over).

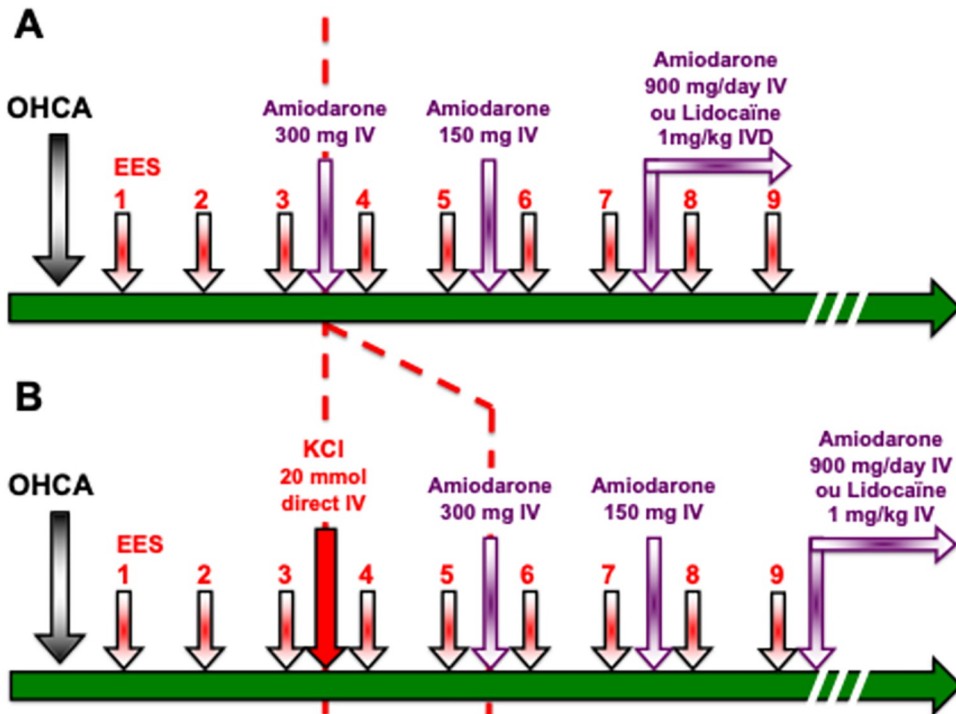

**Fig 1.** Management algorithm for a patient with out of hospital cardiac arrest (OHCA) by VF according to international recommendations (Fig 1A) and in the context of this research (Fig 1B). Epinephrine injections are not shown in the figure for simplicity.

- Patient suffering from an outpatient cardiac arrest of presumed cardiac origin based on clinical history and anamnestic elements, presenting refractory ventricular fibrillation despite 3 external electric shocks.

- Patient with health insurance plan.

## Non-inclusion criteria

- Proven pregnancy.

- Major incompetent (patient under guardianship or curatorship).

- Patient not yet having a functional venous pathway after the 3 external electric shocks.

## Sample size

The experimental plan chosen is a Simon's two-stage optimal design, a type of 2 steps cohort design study. This procedure allows to test the following hypotheses:

- H0: $p \leq p0$ (null hypothesis: insufficient efficiency rate of the KCl treatment)

- H1: $p > p0$ (alternative hypothesis: sufficient efficiency rate of treatment with KCl)

   The probability p0 represents the theoretical efficacy rate, e.g., patient's survival at hospital admission, below which the experimental treatment is considered uninteresting. We must also set a theoretical efficiency rate that we absolutely want to demonstrate, if it exists (p1 or target response rate).

To demonstrate an efficacy of 50% (p1) with a minimum efficacy of 35% (p0, observed survival of the amiodarone group in Kudenchuk [6]), it is necessary to include 81 patients, with an alpha risk of 10% and a power of 90%.

34 patients will be included in the 1st stage. If 12 or less successes are observed, the trial will not be continued and KCl will be found to be ineffective. If at least 13 successes are observed, the trial will continue with the inclusion of 47 new patients in step 2. The treatment will be considered effective, according to these hypotheses, if at least 34 successes are observed in total out of the 81 patients included.

## Statistical analysis

First, a descriptive analysis of the clinical characteristics of the patients at inclusion will be performed. Quantitative data will be expressed as mean ± standard deviation or median [range], and as counts and percentages for qualitative data.

The primary endpoint is the patient's survival at hospital admission defined by the return of an efficient spontaneous cardiac activity. The percentage of successful patients will be calculated at the end of step 1 if the number of successes is not compatible with continuing the study or at the end of step 2 otherwise. The 95% confidence interval of this percentage will be estimated from an exact binomial distribution.

Tolerance data will be described in terms of frequencies and percentages of occurrence.

Likewise, the quantitative secondary endpoints will be described as mean ± standard deviation or median [range], and other qualitative secondary endpoints in the form of frequencies and percentages.

The survival of patients at 3 months can be described globally using Kaplan-Meier curves.

## Investigation centers

Patients presenting with refractory OHCA due to VF refractory to 3 EES will be recruited by the mobile intensive care unit physicians of the following centers:

- SAMU 75—SMUR Necker, Universitary Hospital Necker Enfants—Malades Hospital, Assistance Publique Hôpitaux de Paris, Paris, France;

- SAMU 75—SMUR Pitié Salpêtrière, Universitary Hospital Pitié-Salpêtrière, Assistance Publique Hôpitaux de Paris, Paris, France;

- SAMU 94—SMUR Créteil, Universitary Hospital Henri Mondor, Assistance Publique Hôpitaux de Paris, Créteil, France;

- Paris Fire Brigade, Paris, France.

In hospital care will be performed in one of Paris area according to the French prehospital emergency medical service organization [17].

## Randomization

No randomization is foreseen by the study design patients will be included successively in each investigating center.

## Consent to participate to the study

Considering OHCA as an immediate life-threatening emergency, waiver of informed consent has been approved for this research by the ethical committee (article L1122-1–2 of the French Public Health Code).

### Authorizations and registrations

ClinicalTrials.gov Identifier: NCT04316611, registered in March 2020.

Internal Identification AP-HP180577 / N° EUDRACT: 2019-002544-24.

### Primary outcome measure

The primary outcome is the survival at hospital admission defined by the return of an effective spontaneous cardiac activity at hospital admission.

### Secondary outcome measures

Secondary outcomes criteria are:

- Pre-hospital return of spontaneous circulation (ROSC)

- Time to pre-hospital ROSC

- Total pre-hospital epinephrine dose

- Total number of prehospital EES

- Total number of persistent or recurrent shockable rhythm disorders

- Hemodynamic parameters (heart rate and blood pressure) at hospital admission.

- Survival with good neurologic prognostic (Cerebral Performance Score of 1 or 2) at hospital discharge

- Survival with good neurologic prognostic (Cerebral Performance Score of 1 or 2) at 3 months.

### Duration of the research

The duration of treatment for each subject is a few seconds (time to complete the direct IV 20 mmol KCl injection).

The expected duration of patient participation is 3 months.
The expected length of the inclusion period is 18 months.
The total expected duration of the research is 21 months.

### Trial status

The trial is currently not in the recruitment phase.

### Ethics approval and consent to participate

Considering out-of-hospital cardiac arrest as an immediate life-threatening emergency, waiver of informed consent has been approved for this research by the ethical committee (article L1122-1–2 of the French Public Health Code): committee for the protection of individuals south-east–France (CPP 2019-100-PP).

## Results

No patient is currently included in the study.

## Discussion

The direct IV injection of 20 mmol KCl in a patient presenting with VF, refractory to 3 EES, could result in a rapid return to a spontaneous hemodynamically efficient heart rate.

The effectiveness of KCl to interrupt refractory VF is indeed linked to the peak of hyperkalaemia, and it has been shown that the serum potassium measured 10 minutes after this injection is at the limit of normal (5.5 mmol/l) and normal after 20 minutes. There is therefore no risk of a persistent cardiovascular depressant effect once the peak of hyperkalaemia has subsided. In addition, the 2nd study by Almdahl et al., admittedly in a post-operative context of cardiac surgery under ECC, clearly showed the absence of specific morbidity and mortality in the 1,877 patients who presented VF successfully converted by an injection of 20 mmol of KCl.

In contrast, amiodarone and lidocaine, reference molecules according to international recommendations for interrupting refractory VF, are characterized by cardiovascular depressant effects, which persist for a while after their direct intravenous administration.

The injection of potassium chloride IVD, instead of amiodarone or lidocaine, could therefore allow, like these last 2 molecules, to immediately interrupt refractory VF, but on the other hand without presenting the risk of a persistent cardiovascular depressant effect.

The possible risk of this direct IV injection of KCl can be considered as zero, since the patient is already in refractory cardiac arrest, under specialized cardiopulmonary resuscitation, therefore with an external cardiac massage making it possible to compensate for the absence of spontaneous cardiac activity, and manual or mechanical artificial ventilation to compensate for the lack of spontaneous ventilation.

The possible efficacy of KCl injection in the treatment of OHCA with refractory VF would therefore be likely to improve the prognosis of patients with refractory VF during OHCA. Logically, this should therefore result in a modification of international recommendations concerning this very bad prognosis pathology.

## Supporting information

**S1 Checklist. SPIRIT 2013 checklist: Recommended items to address in a clinical trial protocol and related documents*.**
(PDF)

**S1 File.**
(DOC)

**S2 File.**
(DOC)

## Acknowledgments

All members of prehospital emergency system and in-hospital team taking care of the patients included in the study.

## Author Contributions

**Conceptualization:** Romain Jouffroy, Patrick Ecollan, Bertrand Prunet, Benoit Vivien.

**Funding acquisition:** Benoit Vivien.

**Methodology:** Romain Jouffroy, Caroline Elie, Jean-Marc Treluyer, Benoit Vivien.

**Supervision:** Benoit Vivien.

**Validation:** Patrick Ecollan, Charlotte Chollet-Xemard, Bertrand Prunet, Benoit Vivien.

**Visualization:** Romain Jouffroy, Benoit Vivien.

**Writing – original draft:** Romain Jouffroy, Benoit Vivien.

**Writing – review & editing:** Romain Jouffroy, Patrick Ecollan, Charlotte Chollet-Xemard, Bertrand Prunet, Caroline Elie, Jean-Marc Treluyer, Benoit Vivien.

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
