## [Decision Letter · Decision Letter 0]

26 Sep 2022

PONE-D-22-11622Evaluation of the effectiveness of potassium chloride in the management of out-of-hospital cardiac arrest by refractory ventricular fibrillation: the POTACREH StudyPLOS ONE

Dear Dr. Jouffroy,

Thank you for submitting your manuscript to PLOS ONE. After careful consideration, we feel that it has merit but does not fully meet PLOS ONE’s publication criteria as it currently stands. Therefore, we invite you to submit a revised version of the manuscript that addresses the points raised during the review process.

We look forward to receiving your revised manuscript.

Kind regards,

Yoshihiro Fukumoto

Academic Editor

PLOS ONE

Journal Requirements:

2. During your revisions, please note that a simple title correction is required: please add 'study protocol' to your title, i.e. Evaluation of the effectiveness of potassium chloride in the management of out-of hospital cardiac arrest by refractory ventricular fibrillation: study protocol of the POTACREH Study. Please ensure this is updated in the manuscript file and the online submission information.

"The funders had and will not have a role in study design, data collection and analysis, decision to publish, or preparation of the manuscript."

4. Thank you for stating the following in the Funding Section of your manuscript: 

"The « POTACREH » trial is funded by the French Ministry of Health (Ministère des Solidarités et de la Santé) through the 2018 Inter Regional Clinical Research Program (Programme Hospitalier de Recherche Clinique Inter Régional). The funder has no influence on the recruitment and on care delivered to patients."

"The funders had and will not have a role in study design, data collection and analysis, decision to publish, or preparation of the manuscript."

Reviewers' comments:

Reviewer's Responses to Questions

**Comments to the Author**

1. Does the manuscript provide a valid rationale for the proposed study, with clearly identified and justified research questions?

Reviewer #1: Yes

Reviewer #2: Yes

2. Is the protocol technically sound and planned in a manner that will lead to a meaningful outcome and allow testing the stated hypotheses?

Reviewer #1: Yes

Reviewer #2: Yes

3. Is the methodology feasible and described in sufficient detail to allow the work to be replicable?

Reviewer #1: Yes

Reviewer #2: Yes

4. Have the authors described where all data underlying the findings will be made available when the study is complete?

Reviewer #1: No

Reviewer #2: Yes

5. Is the manuscript presented in an intelligible fashion and written in standard English?

Reviewer #1: Yes

Reviewer #2: Yes

6. Review Comments to the Author

You may also provide optional suggestions and comments to authors that they might find helpful in planning their study.

Reviewer #1: The study protocol aims to address an important clinical question and the use of KCl in refractory VF is a novel treatment. The rationale and motivation of the study is well described in the introduction. I have a few detailed comments:

1. In the methodology section, the study setting is not specified.

2. In the sample size calculation, the outcome is not specified. what did you mean by "efficacy rate"? is that survival outcome? ideally, the sample size calculation should be based on the primary outcome. If this is a preliminary feasibility study, I suggest to change primary outcome to some more intermediate outcome, which requires less sample size.

3. The target population of the study is OHCA, but your participating centers are hospitals. How can you know which hospital will a patient be sent to when the OHCA is first witnessed.

4. The study design should be explicitly described. is that a pre-and post-implementation study. Or paralell cohort study?

Reviewer #2: First of all, this paper is the protocol of future study to evaluate the effectiveness of potassium chloride in the management of out-of-hospital cardiac arrest by refractory ventricular fibrillation.

The Reviewer felt that the current manuscript has somewhat value in terms of the detailed impact of potassium chloride in the management of out-of-hospital cardiac arrest. The Reviewer would like to raise some comments. However, the reviewer sincerely hope that those would be helpful for the authors.

Adjusting for sample size is the most important factor for the success of clinical studies. The authors described in this paper that it is necessary to include 81 patients to demonstrate an efficacy of 50% with a minimum efficacy of 35%. The reviewer wonders the sample number is too small for this study. The causes of OHCA will be multiple, although patients with presumed cardiac outpatient cardiac arrest are included.

How is the cause of OHCA presumed to be of cardiac origin?

7. PLOS authors have the option to publish the peer review history of their article (what does this mean?). If published, this will include your full peer review and any attached files.

Reviewer #1: **Yes: **zhongheng zhang

Reviewer #2: No

---

## [Author Response · Author response to Decision Letter 0]

30 Sep 2022

Dear Editor,

We thank the reviewers for their work and their very interesting remarks.

Please consider the revised version of our manuscript.

Please find below the answers to the reviewers’ comments.

All substantial modifications appear in red and bold in the revised manuscript.

Sincerely yours.

Romain Jouffroy

Journal Requirements:

Answer: the revised version of the manuscript meets PLOS ONE's style requirements

2. During your revisions, please note that a simple title correction is required: please add 'study protocol' to your title, i.e. Evaluation of the effectiveness of potassium chloride in the management of out-of hospital cardiac arrest by refractory ventricular fibrillation: study protocol of the POTACREH Study. Please ensure this is updated in the manuscript file and the online submission information.

Answer: in the revised version of the manuscript, the title was modified accordingly in the manuscript file and on the online submission.

"The funders had and will not have a role in study design, data collection and analysis, decision to publish, or preparation of the manuscript."

Answer: in the revised version of the manuscript, the requested financial disclosure was added.

Answer: all previous queries were corrected in the revised version of the manuscript and in the cover letter.

4. Thank you for stating the following in the Funding Section of your manuscript: 

"The « POTACREH » trial is funded by the French Ministry of Health (Ministère des Solidarités et de la Santé) through the 2018 Inter Regional Clinical Research Program (Programme Hospitalier de Recherche Clinique Inter Régional). The funder has no influence on the recruitment and on care delivered to patients."

"The funders had and will not have a role in study design, data collection and analysis, decision to publish, or preparation of the manuscript."

Answer: All above requested modifications have been done in the revised version of the manuscript.

Answer: In the revised version of the manuscript, ethics statement appears in the Methods section.

Answer: The reference list has been reviewed and is complete and correct.

Reviewers' comments:

Reviewer's Responses to Questions

Comments to the Author

1. Does the manuscript provide a valid rationale for the proposed study, with clearly identified and justified research questions?

Reviewer #1: Yes

Reviewer #2: Yes

2. Is the protocol technically sound and planned in a manner that will lead to a meaningful outcome and allow testing the stated hypotheses?

Reviewer #1: Yes

Reviewer #2: Yes

3. Is the methodology feasible and described in sufficient detail to allow the work to be replicable?

Reviewer #1: Yes

Reviewer #2: Yes

4. Have the authors described where all data underlying the findings will be made available when the study is complete?

Reviewer #1: No

Reviewer #2: Yes

5. Is the manuscript presented in an intelligible fashion and written in standard English?

Reviewer #1: Yes

Reviewer #2: Yes

Answer: We thank the reviewers for their positive comments.

6. Review Comments to the Author

You may also provide optional suggestions and comments to authors that they might find helpful in planning their study.

Reviewer #1: The study protocol aims to address an important clinical question and the use of KCl in refractory VF is a novel treatment. The rationale and motivation of the study is well described in the introduction. I have a few detailed comments:

1. In the methodology section, the study setting is not specified.

2. In the sample size calculation, the outcome is not specified. what did you mean by "efficacy rate"? is that survival outcome? ideally, the sample size calculation should be based on the primary outcome. If this is a preliminary feasibility study, I suggest to change primary outcome to some more intermediate outcome, which requires less sample size.

Answer: We agree and thank the reviewer for this remark. The efficacy rate is patient’s survival at hospital admission. 

3. The target population of the study is OHCA, but your participating centers are hospitals. How can you know which hospital will a patient be sent to when the OHCA is first witnessed.

Answer: We thank the reviewer for this question. As specified in the manuscript, investigation centers are mobile intensive care unit of the 4 centers. The mobile intensive care unit does not always transport the patient to his own centre. A sentence was added in the revised version of the manuscript to clarify this point.

4. The study design should be explicitly described. is that a pre-and post-implementation study. Or paralell cohort study? 

Answer: We thank and agree with the reviewer for this comment. The study design is 2 steps cohort design. A sentence was added in the revised version to clarify this point.

Reviewer #2: First of all, this paper is the protocol of future study to evaluate the effectiveness of potassium chloride in the management of out-of-hospital cardiac arrest by refractory ventricular fibrillation.

The Reviewer felt that the current manuscript has somewhat value in terms of the detailed impact of potassium chloride in the management of out-of-hospital cardiac arrest. The Reviewer would like to raise some comments. However, the reviewer sincerely hope that those would be helpful for the authors.

Adjusting for sample size is the most important factor for the success of clinical studies. The authors described in this paper that it is necessary to include 81 patients to demonstrate an efficacy of 50% with a minimum efficacy of 35%. The reviewer wonders the sample number is too small for this study. The causes of OHCA will be multiple, although patients with presumed cardiac outpatient cardiac arrest are included.

Answer: We thank and agree with reviewer with this interesting remark. However, the Simon’s two-stage experimental design was chosen to avoid unnecessary inclusion and to optimize the number of patients to include in the second phase of the study. This design allows to define the optimal number to treat in the second phase.

How is the cause of OHCA presumed to be of cardiac origin?

Answer: The presumed cardiac origin of OHCA is based on clinical history, anamnestic elements. A sentence was added in the revised version of the manuscript to clarify this.

7. PLOS authors have the option to publish the peer review history of their article (what does this mean?). If published, this will include your full peer review and any attached files.

Do you want your identity to be public for this peer review? For information about this choice, including consent withdrawal, please see our Privacy Policy.

Reviewer #1: Yes: zhongheng zhang

Reviewer #2: No

---

## [Decision Letter · Decision Letter 1]

3 Apr 2023

Evaluation of the effectiveness of potassium chloride in the management of out-of hospital cardiac arrest by refractory ventricular fibrillation: study protocol of the POTACREH Study.

PONE-D-22-11622R1

Dear Dr. Jouffroy,

We’re pleased to inform you that your manuscript has been judged scientifically suitable for publication and will be formally accepted for publication once it meets all outstanding technical requirements.

Kind regards,

Yoshihiro Fukumoto

Academic Editor

PLOS ONE

Additional Editor Comments (optional):

Reviewers' comments:

Reviewer's Responses to Questions

**Comments to the Author**

1. Does the manuscript provide a valid rationale for the proposed study, with clearly identified and justified research questions?

Reviewer #1: Yes

Reviewer #2: Yes

2. Is the protocol technically sound and planned in a manner that will lead to a meaningful outcome and allow testing the stated hypotheses?

Reviewer #1: Yes

Reviewer #2: Yes

3. Is the methodology feasible and described in sufficient detail to allow the work to be replicable?

Reviewer #1: Yes

Reviewer #2: Yes

4. Have the authors described where all data underlying the findings will be made available when the study is complete?

Reviewer #1: No

Reviewer #2: Yes

5. Is the manuscript presented in an intelligible fashion and written in standard English?

Reviewer #1: Yes

Reviewer #2: Yes

6. Review Comments to the Author

You may also provide optional suggestions and comments to authors that they might find helpful in planning their study.

Reviewer #1: My previous comments are well addressed. this study protocol can help to make the study explicitly available.

Reviewer #2: The revised manuscript has been improved. The reviewer has no more comments for the revised manuscript.

7. PLOS authors have the option to publish the peer review history of their article (what does this mean?). If published, this will include your full peer review and any attached files.

Reviewer #1: **Yes: **zhongheng zhang

Reviewer #2: No

---

## [Editor Report · Acceptance letter]

4 Apr 2023

PONE-D-22-11622R1 

Evaluation of the effectiveness of potassium chloride in the management of out-of hospital cardiac arrest by refractory ventricular fibrillation: study protocol of the POTACREH Study 

Dear Dr. Jouffroy:

I'm pleased to inform you that your manuscript has been deemed suitable for publication in PLOS ONE. Congratulations! Your manuscript is now with our production department. 

Kind regards, 

on behalf of

Dr. Yoshihiro Fukumoto 

Academic Editor

PLOS ONE